# Versatile Confocal Raman Imaging Microscope Built from Off-the-Shelf Opto-Mechanical Components

**DOI:** 10.3390/s222410013

**Published:** 2022-12-19

**Authors:** Deseada Diaz Barrero, Genrich Zeller, Magnus Schlösser, Beate Bornschein, Helmut H. Telle

**Affiliations:** 1Departamento de Química Física Aplicada, Universidad Autónoma de Madrid, Campus de Cantoblanco, 28049 Madrid, Spain; 2Tritium Laboratory Karlsruhe (TLK), Institute for Astroparticle Physics (IAP), Karlsruhe Institute of Technology (KIT), Hermann-von-Helmholtz-Platz 1, 76344 Eggenstein-Leopoldshafen, Germany

**Keywords:** confocal Raman microscopy, hyper-spectral imaging, graphene reference samples

## Abstract

Confocal Raman microscopic (CRM) imaging has evolved to become a key tool for spatially resolved, compositional analysis and imaging, down to the μm-scale, and nowadays one may choose between numerous commercial instruments. That notwithstanding, situations may arise which exclude the use of a commercial instrument, e.g., if the analysis involves toxic or radioactive samples/environments; one may not wish to render an expensive instrument unusable for other uses, due to contamination. Therefore, custom-designed CRM instrumentation—being adaptable to hazardous conditions and providing operational flexibility—may be beneficial. Here, we describe a CRM setup, which is constructed nearly in its entirety from off-the-shelf optomechanical and optical components. The original aim was to develop a CRM suitable for the investigation of samples exposed to tritium. For increased flexibility, the CRM system incorporates optical fiber coupling to both the Raman excitation laser and the spectrometer. Lateral raster scans and axial profiling of samples are facilitated by the use of a motorized xyz-translation assembly. Besides the description of the construction and alignment of the CRM system, we also provide (i) the experimental evaluation of system performance (such as, e.g., spatial resolution) and (ii) examples of Raman raster maps and axial profiles of selected thin-film samples (such as, e.g., graphene sheets).

## 1. Introduction

Confocal (laser scanning) microscopy has become the benchmark in many modern microscopy applications, specifically allowing for very high spatial resolution and enhanced contrast in the microscopic images by means of using a spatial pinhole, in order to block out-of-focus light in image formation. For sufficiently transparent samples it is possible as well to acquire depth profiles to construct 3D image maps of thin samples.

In the most common form of confocal microscopy, fluorescence signals—triggered by the scanning laser radiation—are recorded. One specific instrumentation variant adds the capability of Raman spectroscopy, commonly addressed as confocal Raman microscopy (CRM). CRM offers several advantages over other spectroscopic or visible microscopy techniques. First and foremost, Raman spectroscopy/microscopy can be used to analyze small sample areas (of the order of a few μm^2^, or even less) for their full chemical composition. Second, combined with a motorized sample stage, using Raman microscopy, one can generate compound-specific image maps of a sample in three dimensions, applying planar (x–y-direction) and depth (z-direction) scans. Third, chemical evolution of a sample can be followed using time series of Raman spectra.

The principles of CRM and its wide range of applications have been subject to far in excess of 1000 scientific publications, as well as to quite a few review articles (see, e.g., Stewart et al. [1], Opilik et al. [2], and Ni et al. [3], to name but a few). Also, the topics are extensively covered in the recent books by Toporski et al. [4] and Rzhevskii [5]; in the latter, a very useful summary and comparison of a number of commercially available CRM instruments is given. This raises the question: why would one embark on constructing a CRM from scratch if one could simply buy one of the numerous, versatile systems on sale?

Full CRMs, or Raman spectroscopy custom add-ons to existing microscopes—or vice versa—have been built in-house in the past for reasons of cost (commercial CRM systems can have price tags of up to a few €100 k), or because of very specific applications for which commercial instruments were not available, or unsuitable.

In recent years, custom-construction of (confocal) microscopes has made use of “cage” system elements, like those from the *Thorlabs Inc.* collection, which allows for very flexible and versatile setups. Such from-scratch systems include, for example, a modular low-cost epi-fluorescence upright microscope (see Ref. [6]); a microscope for scanning two-photon microscopy “… *that is modular and readily adaptable* …” (see Ref. [7]); a modular scanning confocal microscope “… with digital image processing …” (see Ref. [8]); and a Raman microscope with a motorized sample stage (see Ref. [9]). None of these directly replicate a CRM system in the true sense of the definition; however, the overall idea of a rigid cage system with common subgroups remains the same.

Here, we describe a CRM setup, predominantly built from *Thorlabs*’ optomechanical 30 mm cage elements and associated optical components, aiming at applications (i) in which the sample is of radioactive or toxic nature; and (ii) for which samples ideally can be measured in situ in a hostile environment. Because of these aims, commercial CRM systems are not necessarily suitable, both from the point of view of structural access, and the desire not to expose an expensive instrument unnecessarily to adverse environmental conditions. In particular, our system is designed to potentially probe samples exposed to or loaded with (radioactive) tritium.

Although this latter aspect is not yet fully implemented, proof-of-principle measurements have already been performed on selected samples. Some of these measurements are described in this publication; they demonstrate the suitability of our CRM system for the intended tasks, namely: (i) that surface areas of up to a few cm^2^ could be scanned with spatial resolution of a few micrometers (the aim was to achieve at least <10 μm); (ii) that overall acquisition times per measurement position were of the order 60 s, ideally less; and (iii) that the sensitivity was sufficiently high to yield Raman spectral maps to trace spatial variations or inhomogeneities in the distribution of chemical constituents in thin-films (down to dimension of a molecular monolayer).

The ample technical and experimental information contained in this publication is structured as the following.

Section 2—contains the conceptual description of the CRM setup, together with a detailed description of the construction of its key segments, as well as a brief summary of the motorized sample stage(s).

Section 3—includes a description of alignment optimization for the laser beam passage through the CRM (in commercial instruments this is often already factory-preset); a description of how to realize optimum focusing of the laser beam on the target surface; and the outline of a procedure to automatically keep the sample surface in focus during large-area raster scans.

Section 4—provides a summary of the results of a range of test measurements, to characterize key features of our CRM instrument, and to demonstrate its functionality for the measurement and analysis of thin-film samples. Included are: (i) brief conceptual descriptions of lateral raster scans and axial profiling, together with multi-line Raman spectrochemical analysis data from the related demonstration measurements; and (iii) test results for raster scans of graphene sheets and graphene sensing devices (GFETs), demonstrating the suitability of our CRM for the analysis of monolayer samples.

Appendix A—while all key aspects for the setup and use of our CRM system are covered in the main manuscript, we summarized a selection of useful, additional characterization and test measurement data. Specifically, (i) we added some in-depth procedural details on initial system alignment; (ii) we expanded on the multi-method determination of the laser focal beam diameter on target; and (iii) we provided a comparison of Raman raster maps with images obtained by complementary techniques (here white-light and SEM images), to demonstrate that spectral and structural features could reliably be correlated.

## 2. Experimental Setup

In this section, we describe the general concept of our CRM system, as well as the particular choices for construction components. It has to be stated at the outset that, in general, microscope systems built in-house—like the one described here and those in the aforementioned publications—do not necessarily possess *a priori* the precision and rigidity of commercial instruments. 

As mentioned in the introduction, one key aspect for engaging in the construction of a dedicated CRM system was the intended application of investigating samples exposed to tritium; or in situ monitoring of samples placed within a tritium atmosphere, in order to trace chemical changes. Therefore, it was paramount the CRM could handle such scenarios without compromising the whole system to become contaminated. 

In this context, strenuous efforts have been made to come up with solutions, which, as much as possible, replicate the construction concepts and performance criteria of commercial confocal (Raman) microscope setups but at the same time, providing the flexibility for adaptation to investigate the aforementioned types of samples in a handling environment such as, e.g., a glove box. Note that, because of the (slightly) reduced precision tolerance in our in-house built CRM, several positional adjustment and alignment aids—beyond those normally encountered in commercial instruments—have been incorporated as well. Note also that for reasons of system rigidity and potential setup within one of our TLK glovebox environments, the CRM is set up horizontally on a breadboard. Note however that, in principle, the complete CRM-on-breadboard system can be arranged in any orientation (including in an upright configuration, as in standard microscopes, if needed).

The overall concept of our CRM system is shown in Figure 1 and Figure 2 (schematic drawing and overview photo, respectively). As in commercial CRM realizations, the overall system can be sub-divided into a number of key functional groups. In our case, these comprise the following three system sub-groups:-Segment A. Laser light coupling into the CRM via a single-mode (SLM) optical fiber; (optional) monitoring of the laser power; and guidance of laser and Raman light through the microscope objective. Note the subtle differences in Figure 1 and Figure 2; the schematic sketch in Figure 1 shows the setup of our second CRM, in which the laser coupling was switched to the opposite side as in the original CRM (photo in Figure 2), making the system more compact.-Segment B. Wide-field imaging arm to record images of the sample using a 2D CMOS camera.-Segment C. Confocal detection arm to image the laser excitation region on the sample onto the confocal pinhole and onto a fiber bundle carrying the Raman light to the spectrometer.

In addition, a separate unit comprises the sample holder, which is mounted on a motorized precision xyz-translation assembly; details of this system group are not shown in the schematic overview in Figure 1 but are described in Section 2.2.

The individual CRM segments are described in more detail in the subsections below, highlighting specific design details (including the specifications for key components of the setup.

### 2.1. CRM Construction

As is evident from Figure 1 and Figure 2, the concept of construction of the CRM relies largely on readily available off-the-shelf components, which on the one hand allow for sturdy construction, but on the other hand provide as much flexibility as possible in the setup (to be adaptable to different application conditions). For reasons of compatibility and easy extension, the setup is based on optomechanical 30 mm cage elements (*Thorlabs*), with matching optical components—only the Raman spectral filters are sourced from a different supplier (*Semrock*).

To maintain rigidity of the CRM, it is mounted horizontally on a dedicated optical breadboard, as mentioned earlier. Details of the three building blocks (Segments A to C, with reference to Figure 1) are described below; a short summary of component functionalities and commercial sources of key elements is collated in Table 1.

**Laser coupling, Raman light filtering, and microscope objective—Segment A**. For flexibility of use in potentially toxic or radioactive environments, the Raman excitation laser is coupled into the CRM by using a single-mode FC/PC or FC/APC fiber optic patch cable. The core diameter of common single-mode optical fibers used for transmission of 532 nm laser radiation (like e.g., the *Thorlabs* SM450 type) exhibit a mode-field diameter of the order of MFD_SM450_ ~ 3.5 μm. It thus acts equivalently to the pinhole used in common free-space coupling of the laser radiation into a confocal microscope (see, e.g., References [10,11]). When connecting this fiber to a fixed-focus collimation package (here we use a *Thorlabs* F260FC-A collimator) a collimated laser beam is generated of about 2 mm in diameter. 

Note that this configuration does not require a complex fiber-launch assembly with the associated need for alignment. The only alignment requirement is beam positioning along the optical axis of the CRM cage system, which can easily be implemented using a $\left(x,y,\theta_{x},\theta_{y}\right)$-kinematic mount (e.g., KC1-S/M, *Thorlabs*). It should be noted that as an alternative to the standard single-mode fiber, as discussed here, polarization-maintaining single-mode fiber types are also suitable.

Before entering the actual CRM-structure, the collimated laser beam passes through a laser line filter (LL01-532-12.5, *Semrock*) to remove any fluorescence and stimulated Raman light, which might have been generated in the fiber.

The 532 nm laser beam is finally directed to the microscope objective (and on to the sample) via a single-edge dichroic beam splitter (Di02-R532-25×36, *Semrock*). This beam splitter exhibits reflectivity for the laser light of R_532 nm_ ~ 95% and transmission for the Raman light of T_Raman_ > 93%.

The microscope objective used for the current test measurements, and for intended future application of Raman imaging in hostile environments, is a low-cost infinity-corrected 10× objective (LIO-10X, *Newport*/*MKS*). This choice was made for two reasons; namely (i) to minimize the cost for an optical component, which may need replacement after exposure to a toxic or radioactive sample/environment; and (ii) because relatively large areas were to be raster-scanned, which with larger-NA objectives (i.e., 20× objectives, or higher) may result in excessively long scan durations. Of course, any other standard RMS-threaded objectives may be used, in principle, which are suitable for mounting in the cage setup.

As an ancillary measurement device (in Segment A), a laser power monitor is incorporated in the form of a Si-photodiode (SM1PD1A, *Thorlabs*). For this, one exploits that the dichroic beam splitter transmits ~0.5% of the 532 nm radiation; even for relatively low laser power (in our case in the range 5–100 mW), this is quite sufficient to be detectable. The photodiode is mounted with an angle of a few degrees, to avoid back-reflection from its entrance window into the CRM. Note that a diaphragm fitted to the dichroic beam splitter cube blocks any light reflection from the photodiode from entering the cube, thus avoiding additional scattered-light background.

**The optical imaging arm—Segment B**. The light emerging from the sample is split in the ratio 10:90 (R:T = reflection:transmission) by a non-polarizing beam splitter cube (BS025, *Thorlabs*). The reflected light is focused by an achromatic lens of focal length *f*_acr_ = 150 mm onto a color CMOS camera (CS165CU/M, *Thorlabs*), with an imaging area of 1440 × 1080 pixel (about 5.0 × 3.7 mm^2^) and pixel size 3.45 × 3.45 μm^2^. 

Note that the focusing is equal to that in the Raman detection arm of the CRM, so that direct comparison between wide-field (white-light) and confocal Raman image dimensions can be made. Note also that in the case that one wishes to record the laser beam focus to be imaged with the camera, a neutral density filter may need to be inserted in the optical imaging arm, to avoid saturation of the CMOS camera sensor. 

**The confocal detection arm—Segment C**. After having passed through the beam splitter cube (which extracts 10% of the light for the wide-field camera imaging), the Raman light has to be focused onto the confocal pinhole. 

In order to remove any residual 532 nm laser light, a RazorEdge^®^ long-pass edge filter (LP03-532RU-25, *Semrock*) is incorporated into the Raman light pass prior to the focusing lens. Although in principle the suppressing action of the single-edge dichroic beam splitter (Di02-R532-25×36, *Semrock*)—used for the laser coupling into the CRM—should suffice, its transmission for laser light is of the order of 0.5% (see the transmission data in the spec sheet of the Di02-R532 beam splitter). When using fiber coupling between a Raman light source and the spectrometer, as is the case here, residual laser light might already be sufficient to generate unwanted Raman signals and fluorescence in the fiber, potentially interfering with the Raman light from the CRM sample.

The Raman light is focused onto the confocal pinhole by an achromatic lens of focal length *f*_acr_ = 150 mm. The lens position can be adjusted both in a lateral and axial direction. With the former adjustment, one maintains on-axis light propagation within the cage assembly; with the latter, exact focusing onto the pinhole can be achieved (note that the pinhole is mounted at a fixed axial position).

The pinhole can be exchanged to adapt its diameter to the optimum associated with the specific confocality conditions—stemming from (i) the laser focal beam diameter on target; (ii) the objective’s focal length and numerical aperture; and (iii) the focal length of the achromatic lens. The optimum for our current CRM configuration is in the range 75–100 μm, as determined from the experimental data described in the Appendix A. 

Most of the data discussed in the results sections were recorded using either of two standard pinholes of diameter 75 μm and 100 μm (P75K, P100K, *Thorlabs*). In order to confirm the expected behavior of the Raman signal, with respect to axial and lateral position/dimension, for comparison, we also used smaller pinholes of diameter 50 μm and 25 μm (P50K, P25K, *Thorlabs*); see the remarks in Section 4.3.

Since the pinhole is mounted in an anodized aluminum housing, which subsequently is inserted into a SM1 lens-tube adapter, its position with respect to the cage optical axis may vary by several micrometers. Therefore, the pinhole assembly is inserted into a precision xy-translation stage, compatible with the 30 mm cage (ST1XY-S/M, *Thorlabs*), to allow for optimal alignment of the pinhole to the focal image from the achromatic lens.

The Raman light, passing through the pinhole, is collected by a custom-fabricated optical fiber bundle (*CeramOptec*). It comprises 48 individual fibers of core diameter 100 μm, which are arranged in a circular fashion at the CRM end and a line-pattern at the spectrometer end (matched to the height of the spectrometer’s CCD detector). The circular-end diameter of the bundle is 1 mm. In order to illuminate all individual fibers of the bundle, its position can be adjusted in both a lateral and axial position, in accordance with the numerical aperture of the light passing through the pinhole. 

Note that the setup provides the flexibility to use different fiber bundles or single-core multimode fibers, in the case that the spectrometer system is different to the one used here (iHR 320 imaging spectrometer with a Syncerity FI-UV-VIS CCD-array detector, *Horiba*).

### 2.2. Motorized Sample Motion

As was pointed out earlier, in the CRM described here, the raster scanning of samples is based on spatially moving the sample stage rather than utilizing a CRM-internal laser beam scanner. For the intended applications, two main aspects were important. 

First, the CRM should possess the capability that (i) Raman probing and imaging could be performed across the full sample area of up to 1 cm^2^, and that (ii) in any selected location within this area, the sample could be held in focus, i.e., within the Rayleigh range; see Appendix A for further details.

Second, good spatial localization/raster stepping and reproducibility should be achievable, matched to the actual laser focal beam diameter (FBD). Note that for the mostly-used 10× objective, this is FBD ≈ 7.2 μm (equivalent to FWHM ≈ 4.2 μm); for further details, see Section 4.2 and the Appendix A.

Therefore, the positional adjustment capabilities of the xyz-sample stage should be adjustable with a precision of the order of 2 μm (or better if objectives with high-NA are used). For comments on reproducibility, see the Appendix A. 

In the two CRM systems built within the framework of this study, different motorized xyz-assemblies were tested. These were: -a three-axis combination of stepper-motorized translation stages *Standa* 8MT173-20-MEn1 with optical encoders (with full-step resolution = 1.25 μm and nominal micro-step resolution = 0.156 μm ≡ 1/8 step); and-a combination of piezo-inertia translators; comprising two *Thorlabs* PD1/M linear stages (with nominal step resolution = 1 μm) for the lateral xy-motion, and a *Thorlabs* LX20/M translation stage with PIA25 actuator (with nominal step resolution = 20 nm) for the axial z-translation. 

The important characteristic data for these two assemblies are collated in Table 2.

Great care was taken in the design and construction of the actual sample holder, which was attached to the motorized xyz-assembly. The concept was to ascertain (i) that it was as simple as possible; (ii) that the range of samples anticipated in this and future studies could be mounted reproducibly, and be secured in predefined positions; and (iii) that no damage to a sample was incurred when mounting or removing it. A few points regarding the sample holder design are noteworthy.

First, in contrast to commercial confocal microscopes, the sample stage was not directly integral to the microscope structure, and therefore inherently aligned, but formed a separate building block. This meant that additional $\theta_{x},\theta_{y}$ yaw adjustments were desirable to pre-align the sample surface as precisely as possible in the lateral motion plane. This was accomplished by incorporating a kinematic platform mount (*Thorlabs*).

Second, the sample holder comprises a magnetic quick-release base (*Thorlabs*) which can be mounted/removed from the $\left(\mathrm{xyz},\theta_{x},\theta_{y}\right)$-assembly, with mounting repeatability of the order 10 μrad. In this way, potentially toxic or radioactive samples can be handled within a protective, secure environment prior to the final mounting. The samples themselves are held in position using dedicated template masks, matched to the size and shape of the sample, and fixed with a couple of holding clips.

Of course, the two options described here are not the only ones possible: different motion stages and mounting devices may be utilized, which are designed and adapted to match any specific application (for example, if larger areas are to be scanned or higher positional resolution is required).

Finally, note that the motorized sample holder is mounted on the same breadboard as the CRM structure itself, therefore guaranteeing that the complete system comprises a rigid unit which has proven to be largely immune to the environmental, vibrational noise encountered at the installations in our tritium laboratory. The two CRM implementations described in this publication comfortably fit onto a breadboard of size 500 × 750 mm^2^ and 650 × 750 mm^2^, respectively.

## 3. Alignment and Characterization

Commercial confocal (Raman) microscopes are normally factory-aligned, mostly by the initial, precision engineering design and during instrument manufacture. In general, only minor tweaking may be required, from time to time, to recover optimum alignment and performance. 

This is not necessarily the case for a custom-constructed instrument, which is assembled from individual, kit-type components (for the CRM described in this publication, the *Thorlabs* 30 mm cage system). Thus, the required precision for the optical paths of the order of just a few micrometers may not be achievable *a priori* during the assembly phase. Consequently, such an instrument needs to undergo initial beam path alignment, and to offer the option to realign the system with reasonable ease, in the case (i) that the structure is perturbed (e.g., by shock or thermally induced movements); or (ii) that components need to be exchanged (e.g., different microscope objective, focusing lens, or pinhole).

Thus, for flexible and easy alignment, all critical components are mounted in appropriate adjustment components, to allow for the necessary lateral (xy-translation), axial (z-translation), or angular ($\theta_{x}\theta_{y}$-yaw) movements.

In the following paragraphs, a brief description of alignment procedures is provided, which highlights key steps in the setup of the confocal Raman light path and—related to that—the setup of the wide-field CCD camera arm.

### 3.1. Alignment of Confocal Raman Light Path

The all-important CRM alignment, so that the sample excitation region is imaged precisely onto the confocal pinhole along the optical axis of the system, is normally achieved through factory-preset rather than manual—as just mentioned. Here, we briefly summarize the important sequence of steps necessary to achieve the required alignment for our in-house built CRM. Throughout the procedure, cage targets (CPA1, *Thorlabs*) constituted the central alignment tool. 

In our CRM, a collimated laser beam—originating from the fixed-focus fiber collimator unit in a $\left(x,y,\theta_{x},\theta_{y}\right)$-kinematic mount (see Section 2.1)—is propagating via the dichroic beam splitter through the objective onto the target. Because of the delicate positioning of the beam splitter in its mounting cube (perfect vertical mounting with respect to the optical cage axis was difficult to achieve), alignment of the incoming laser beam constitutes a complex, though straight-forward task. The key steps in the alignment procedure are provided in the Appendix A.

Note that throughout the alignment, a crystalline-Si (c-Si) target—SEM finder grid substrate (EM-Tec FG1, *Micro-to-Nano*)—is used as a target (replicating a sample surface), positioned roughly at the focus of the objective. This particular choice was made for two reasons. First, c-Si exhibits a high reflectivity of ~37% at the laser wavelength $\lambda_{\mathrm{laser}\;}$= 532 nm (see, e.g., References [12,13]); thus, it is suitable to act as a mirror for imaging of the laser focal spot onto the CMOS camera. Second, it has a strong Raman peak at 520.9 cm^−1^ [14] (since the early days of Raman spectroscopy with lasers, this particular sample and its Raman peak are utilized for spectral calibration of Raman microscopes).

Finally, since several of the optical components introduce a (slight) wavelength-dependent directional propagation of the Raman light, fine-tuning of the achromatic lens position with respect to the pinhole needs to be made; for this, the Raman signal from the Si-grid finder target is recorded and its signal amplitude maximized. Note that for optimal illumination of the fiber bundle, which couples the CRM to the spectrometer, its xyz-position needs to be adjusted as well, relative to the pinhole location.

### 3.2. Alignment and Use of the Wide Field CMOS Camera Path

When selecting the same type of lens (*f*_acr_ = 150 mm) for both the confocal Raman and the imaging arms of the CRM, the distance between the CMOS camera sensor and this lens should equal the distance between the pinhole and the focusing lens in the confocal Raman arm (Segment C). Then, information from either image plane may be used in determining whether the sample surface is in focus, or not. 

Note that the alignment of the wide-field imaging arm (Segment B) is less critical than that of the confocal arm (Segment C); basically, only the imaging achromatic lens has to be positioned precisely in its axial location. The required alignment steps are summarized in the Appendix A.

At the end of the procedure, a maximal Si Raman signal will correspond to a minimum-sized laser spot image, i.e., the two arms are now perfectly matched with respect to each other.

### 3.3. Axial Focussing onto the Sample Surface

With the confocal and the wide-field imaging arms aligned and focally matched, the image of the laser spot on the CMOS camera can be used to determine the focus position of any sample, which exhibits decent reflectivity for laser light with $\lambda_{\mathrm{laser}\;}$ = 532 nm. This can be achieved without the need to perform an extended axial scan of the sample, which then is fitted to a Lorentz function to derive the maximum and Rayleigh depth (as outlined in the Appendix A). Such a proposal for rapid (automated) focus control is discussed, e.g., by Yazdanfar and co-workers [15]; note that, automated focus control is frequently used in commercial confocal laser microscopes.

In order to find the minimized laser spot image recorded by the CMOS camera, the sample is simply moved in axial z-direction around the guessed focus location. Using an intelligent algorithm, as suggested in Reference [15], the exact focal position can be determined from only a few images. Thus, it constitutes a much faster procedure than using the aforementioned axial Raman profiling.

Note, however, that for transparent samples/substrates, one has to ensure that the laser spot is indeed on the sample surface and not on the (metal) sample holder surface below. For such transparent samples, one can typically observe two focus positions with minimized laser spot size. The position closer to the CRM objective corresponds to the investigated sample surface (such as the graphene films discussed in Section 4). Note also that for transparent samples, it is recommended to roughen the surface of the sample holder, or “blacken” it, in order to reduce the intensity of its laser light reflection.

Finally, for the generation of planar (xy-direction in our geometry frame) images of a sample, during the scan, the sample surface needs to stay within the Rayleigh range of the laser focus (z-direction). Due to unavoidable offset and rotations in the assembled xyz-motion-stage and uneven mounting in the sample holder, the sample and the laser focus planes are not necessarily parallel to each other on a sub-degree level. Thus, the sample surface would likely move out of focus during a full planar scan. A simple procedure for automatic focus compensation is briefly described in the Appendix A.

## 4. Results of Test Measurements

The confocal Raman (CRM) system described in Section 3 has been tested for performance and reliability prior to routine measurements but using—as far as possible—samples expected in future applications in our laboratories. The actual CRM-setup is complemented by the external functionality groups of (i) the excitation laser; (ii) the Raman spectrometer; as well as (iii) data acquisition and evaluation software.

**The excitation laser.** For the tests discussed in this publication, we have used a medium-power fiber-coupled DPSS laser module (“Matchbox” model 0532L-15B-NI-PT-NF, *Integrated Optics*). This temperature-stabilized module operates at a nominal wavelength of 532 nm (multiple longitudinal modes, with bandwidth Δλ < 0.3 nm and TEM_00_ beam profile), with an output power of ~100 mW. It is directly fiber-coupled with a polarization-maintaining single-mode fiber (PM-SLM), which provides a near-Gaussian beam profile at the fiber exit. The fiber output end is terminated by a standard FC-connector (FC/PC) and is coupled to the CRM entrance collimator (see Section 2.2). 

For our setup with 10× objective, the laser power entering the CRM was limited to ~80 mW, in order not to cause damage to samples by excessive irradiance (by rule of thumb, the power density on target should not exceed ~10^6^ W/cm^2^; see, e.g., Reference [4]). 

Note that as an alternative to the MatchBox laser, we have also successfully implemented free-space coupling of a stand-alone 532 nm DPSS laser, via an input-collimator → SLM fiber → output-collimator chain, in the case that no direct-fiber-coupled laser module is available.

**The Raman spectrometer.** In principle, any spectrometer suitable for Raman spectroscopic detection and analysis is suitable (we have used both Horiba and Acton Research spectrometers, with a variety of CCD array detectors). For the measurements described in this publication, we used an iHR320 (*Horiba*) spectrometer—with 300 grooves/mm grating—and a Syncerity CCD detector (*Horiba*)—array 1024 × 256 pixel, with pixel size 26 × 26 μm^2^. For an entrance slit of ~100 μm, one obtains a spectral resolution of about 0.95 nm (equivalent to ~24.7 cm^−1^). This resolution was sufficient for the demonstration purposes addressed in this publication, but can easily be increased by selecting a grating with higher dispersion and/or narrower entrance slit size, in principle, for future applications. 

**System control and data acquisition**. The spectral data were recorded using our *LabVIEW*™-based analysis software suite. This is a successor to the original LARAsoft package [16], which also was extended by a sub-vi module to synchronize the movement of the xyz-motorized-stage and the data recording with the CCD detector. This enabled fully automated scans of xy-lateral image maps or axial depth profiles (or combinations thereof) of a sample. In general, for mono- or few-atomic layer samples—such as graphene—two spectrum acquisitions per raster point were taken, each normally of a duration of 20 s. Note that for thicker thin-film layers, this acquisition time could be reduced substantially. The CCD array data were sampled by on-chip binning (here, two bins of 128 lines each). This combination of multiple accumulation and multi-bins allowed for easy implementation of dead pixel and cosmic ray removal, utilizing the bespoke routines built into LARAsoft. 

### 4.1. Concepts of Raster Scans and Raman Signal Analysis

The most common procedural sequence in Raman imaging is that for each pixel position in the probed area, a complete Raman spectrum is recorded. This results in the generation of a hyperspectral data cube, i.e., the collection of spectra taken from a 2D XY-area scan is represented by a 3D-data cube, whose “X/Y”-dimensions hold the spatial (scan) information and whose (third) “S”-dimension (or spectral-dimension) is associated with the recorded Raman spectra. Ordinary aerial scans may be expanded to three dimensions, i.e., including depth profiling in the “Z”-direction. Then, the complete spatial–spectral data are represented by a 4D-data cube (four-dimensional = XYZ-spatial + S-spectral). 

However, it is not normally intuitive to visualize and interpret 3D-/4D-array structures of spectral data for a scanned sample. Therefore, in general, image reconstruction software handling the interpretation of such data cubes is utilized. 

In order to unravel molecular compound information from the hyperspectral data cube and to construct spectrochemical maps of the sample, probably the most prominent method is spectral “unmixing”. This means one first decomposes mixed-component spectra into a series of constituent (reference) spectra, and then deduces the spatial abundances from the signal amplitudes in each spectral interval, for each spatial “pixel” position (see, e.g., References [17,18,19]); this procedure is often referred to as “slicing”. Of course, the success of such slicing normally hinges (i) on applying intelligent questioning and/or prior knowledge to/of the sample structure; and (ii) on utilizing additional interpretative methodologies of component-signal correlation (see, e.g., [20,21,22]). 

It should be noted that full hyperspectral imaging analysis was not applied to the graphene examples discussed in this publication (for hyperspectral analysis of graphene see, e.g., [23]). Rather only single- (one slice) or multi-line (here up to three slices) analysis was utilized, which was deemed sufficient to demonstrate various Raman imaging aspects of our CRM.

Throughout this work, a GFET-S10 chip (for sensing applications), or other large-area graphene samples from *Graphenea* were used. Said chip sample provides 36 graphene devices distributed in a grid pattern on the chip. The size and geometry of the individual graphene areas varies from 50 × 10 µm^2^ to 200 × 200 µm^2^. Each of these graphene areas is contacted with 2 or 6 gold connections, for 2-probe or Hall-bar geometry, respectively. 

Basically, the GFET-S10 chip sample allows one—without the need to change the sample—to investigate a range of different properties and effects. In particular, one may (i) distinguish different materials (gold/graphene/SiO_2_); (ii) different material transitions graphene⟷gold, graphene⟷SiO_2_; and (iii) macroscopic graphene regions, which according to the manufacturer’s specifications, are entirely 1LG. 

Furthermore, the chip is similar to the GFET-S11 sample, which is planned to be used in future experiments for Van-der-Pauw resistivity measurements of graphene structures (see, e.g., [24]), to probe changes in the layer subsequent to chemical treatment. Thus, spectroscopic pre-characterization of typical graphene areas—as done here—will provide graphene 1LG reference data.

In the upper-left of Figure 3, the schematic layout of the GFET-S10 sample structure is shown, together with an enlarged view of the 50 × 50 µm^2^ GFET device selected for the demonstration of Raman mapping. A Raman raster map associated with this selected device is shown in the upper-right part of the figure. The scan was carried out with step-increments of ~9 µm, in both x- and y-directions. Overall, an area of ~350 × 350 µm^2^ was scanned, with the graphene layer located approximately in the middle. 

Note that the Raman map was generated from a single slice in the hyperspectral stack at about 1330 cm^−1^. This slice interval of width ~100 cm^−1^ includes the D-line of graphene, while excluding signal contributions from the neighboring graphene G-line and the SiO_2_ 2TO-line. The associated (integrated) intensity data were normalized to the maximum observed D-line signal. One should note that for the spectra shown in Figure 3, non-Raman background removal was NOT applied, in contrast to the other measurement data discussed later in this publication.

In the lower half of Figure 3, full Raman spectra from the hyperspectral stack are displayed, recorded at the three specific locations marked in the map—P1 is located on the graphene layer; P2 on a gold-contact; and P3 on the SiO_2_ substrate area. If one inspects these spectra at the position of, e.g., the 1330 cm^−1^ slice, it is easy to see as to why one can also distinguish the three structural areas evident in the Raman spectral map: -P1: the spectral slice incorporates the D-line signal of graphene (plus residual background from the substrate material);-P2: there is almost no signal contribution from the SiO_2_ substrate within the slice;-P3: fluorescence from the gold contact contributes to the spectral slice.

Note that the results shown in Figure 3 are only qualitative, to demonstrate the capabilities of our CRM for spectral and structural analysis. Some examples of semi-quantitative spectral analyses are discussed further below. 

Full hyperspectral quantitative data analysis of the stack for GFET and other graphene samples will be presented in a future publication, specifically with the view of distinguishing between mono- (1 LG) and multi-layer (n LG) graphene, and reaction-induced chemical modifications of these layers.

Finally, in addition to the distinction of the different materials and their structural distribution, this wide area raster scan demonstrates that the procedure to keep the sample in focus during a scan (for relevant procedural information, see Appendix A is working well; no significant change in the SiO_2_—or Au-signal intensity across the whole raster scan area was observed.

### 4.2. Determination of the Laser Focal Beam Diameter

Commonly, commercial laser beam profilers are used to capture and analyze the spatial intensity profile of a laser beam in a plane transverse to the beam propagation path; devices based on CCD-array camera techniques with direct illumination of the camera sensor are the most popular. However, such instruments are not necessarily suitable if the beam diameter to be analyzed approaches the pixel size of the camera sensor, mostly of the order of 2–5 μm. For the spot size of a laser beam focused through a microscope objective, this may become insufficient; in addition, it might be physically impossible to arrange the beam profiler in the focal plane of a microscope. 

Therefore, instead of direct beam profiling the (indirect) knife-edge technique is often utilized (see, e.g., [25,26]): a slit or blade cuts the laser beam before detection by a power/intensity detector. By stepwise recording the integral intensity profile for a number of cuts, the original beam profile can be reconstructed, nowadays predominantly using algorithms developed for tomography.

Here, we have applied a slight modification of the standard knife-edge method. Instead of measuring the detector response to the integrated laser intensity, partially obscured by the knife-edge, the laser beam moves across a sharp boundary with a different response to the laser radiation on both sides. In particular, we have exploited the Raman signal response associated with different (molecular) sample compounds.

For the determination of the laser spot size, we utilized a sample with distinct “edges” in its structure, namely a silicon SEM finder grid substrate (*Micro to Nano* EM-Tec FG1).

While primarily designed for SEM applications, the finder grid substrate is equally suitable for reflected light or Raman microscopy. The sample of size 12 × 12 mm^2^ comprises a silicon monocrystal with <100> surface orientation onto which a 1 × 1 mm^2^ grid of chromium (Cr) is deposited (nominally with thickness 75 nm and width 20 μm). The laser beam was focused onto the Si-substrate, minimizing its diameter as observed via the CMOS camera (for details see the Appendix A). Subsequently, the sample position was changed in one direction (“horizontal” scan), in step increments of 0.5 (≡ 0.625 μm) across four consecutive grid lines, both in a forward and backward direction. The scan over one individual Cr-grid line is shown in the right data panel of Figure 4. 

Details of the fit procedure to derive the FBD from the data plot are given in the Appendix A. In the data plot, the fitting curves of the leading (red line) and trailing (green line) edges are included; these fits replicate the experimental data points rather well. From these data fits, the laser focal beam diameter (FBD) can be determined; here, we define a beam diameter as the 1/e^2^ point in the Gaussian TEM_00_ profile function. The average over the eight leading/trailing edges of the four grid lines contained in the scan yields FBD = 7.23(13) μm on target (the statistical error value is linked to the last numerical digit of the average value). The following additional observations can be made, based on the results from this finder grid scan.

First, the spacing between Cr-grid lines was determined experimentally as 800.0 ± 0.2 steps, which is equivalent to 1000.0 ± 0.25 μm; this replicates the nominal grid line spacing within the mechanical accuracy of the stepper driver of the translation stage. 

Second, the experimental width of the Cr-grid lines, as determined from the half-intensity points of the knife-edge scan(s), is found to be w_GL_ = 20.5 ± 0.3 μm. This value is slightly larger (about 2–3%) than the nominal value stated in the data sheet, which is most likely associated with the uncertainty in the analysis methodology.

Third, it is worth noting that the laser spot size on target achieved here does not constitute the ultimate (diffraction) limit. As outlined in Section 2.1, for simplicity of alignment, the laser radiation is fed into the CRM system using a single-mode fiber (*Thorlabs* P1-460B-FC-2) coupled to a fixed-focus fiber-collimator (*Thorlabs* F260FC-A). 

The associated laser beam waist of ~2.8 mm is substantially less than the diameter of the objective’s rear aperture of ~9 mm. As a consequence, the experimental laser beam spot size is quite a bit larger than the theoretical limit; in the case that the full numerical aperture of the objective, NA_obj_, be used for laser illumination (according to the common approximation $w_{0,@\mathrm{sample}}≅\lambda /\left(\pi \cdot {\mathrm{NA}}_{\mathrm{obj}}\right)$, the laser FBD on target would be of the order 2.5 μm).

Finally, we like to point out that the methodology just discussed can be applied to any sample which exhibits a sharp transition between chemically distinct, spatial features. As an example, we have demonstrated this for the GFET sample discussed in Section 4.1, namely for the edge transition between the graphene sheet and the SiO_2_/Si substrate; the results are included in the Appendix A.

### 4.3. The Use of Different Pin Holes and Objectives

The objective and the pinhole are two key elements affecting the performance of a confocal (Raman) microscope. In general, as a good compromise, the pinhole diameter is chosen to equal that of the Airy-disk diameter in the confocal image (Airy-disc diameter ≡ 1 Airy unit [AU]). For the majority of the work discussed in this publication, we used a 10× infinity-corrected objective (with ${\mathrm{NA}}_{\mathrm{obj}}$ = 0.25), although other higher-NA objectives were tested as well.

The size of the Airy-disk ($\emptyset_{\mathrm{Airy}\;\mathrm{disk}}$), and therefore the confocal pinhole ($\emptyset_{\mathrm{confocal}\;\mathrm{pinhole}}$), depends on the numerical aperture of the objective lens (${\mathrm{NA}}_{\mathrm{obj}}$), the wavelength (*λ*) of the imaged light, and any magnification ($M_{\mathrm{CRM}}$) up to the pinhole:(1)$$\emptyset_{\mathrm{confocal}\;\mathrm{pinhole}}\equiv \emptyset_{\mathrm{Airy}\;\mathrm{disk}}\approx 2.44\times \left(\frac{\lambda }{2\cdot {\mathrm{NA}}_{\mathrm{obj}}}\right)\times M_{\mathrm{CRM}}$$

Detailed considerations of the pinhole diameter and its effect on resolution may be found, e.g., in references [27,28,29]. Note however, that this equation is only valid if the full numerical aperture of the objective is exploited. 

In our CRM setup, only a smaller effective numerical aperture, ${\mathrm{NA}}_{\mathrm{obj},\mathrm{eff}\;}$, is utilized. In the Appendix A, we provide a brief outline of how we determined the optimal pinhole diameter which takes into account ${\mathrm{NA}}_{\mathrm{obj},\mathrm{eff}\;}$. Based on those results, most of the data discussed in this publication were recorded using a compromise pinhole of 100 μm diameter, which maximized the Raman light signal without sacrificing confocality.

### 4.4. Raster Scans of Graphene Samples

In order to demonstrate the capabilities of our CRM system for spatially-resolved spectrochemical analysis, we executed a series of raster scans of graphene samples, predominantly GFET devices (*Graphenea*) with different sizes of the graphene area. As pointed out earlier, in principle, the full area of the GFET multi-device substrate of 10 × 10 mm^2^ could be scanned with μm-resolution. However, in the proof-of-principle scans discussed here, in general, raster maps were limited to areas of the order 50 × 50 raster points, in order to keep the total acquisition time to a reasonable duration. Note that two back-to-back acquisitions were taken at each raster point location of the order of 10–20 s duration each (this dual-spectra acquisition is required to apply the LARAsoft-internal cosmic-ray removal procedure [16]). This leads to a total acquisition time for such a 50 × 50 raster map of roughly 14–28 h.

The lateral step increments were adjusted to match the properties of the laser focal beam diameter (here, FBD ~ 5.7 steps, equivalent to ~7 μm, as determined in Section 4.3). For quick survey scans, step increments equal or larger than the FBD were utilized (ΔS = 5–10), while for the purpose of revealing finer details in the sample surface structure step increments of ΔS = 1–2 were usually selected. Recall from Section 2.2 that a full-step increment ΔS = 1 corresponds to a spatial displacement of 1.25 μm.

Because of the inherent backlash in the motorized translation stages, the raster scans were normally executed unidirectional to avoid line-by-line distortions in the Raman images. Note that from forward/backward scans across four sequential Cr-rid lines of the SEM finder grid sample (see Section 4.2), we determined this backlash to be ~1.8 full-steps. Thus, the raster maps constitute consecutive line-scans in a (horizontal) x-direction, returning to the x-reference location after completion; for the subsequent x-direction scan, the sample position was incremented in a (vertical) y-direction by the chosen ΔS.

After completion of the full raster-scan, the hyperspectral data cube was analyzed according to the procedure described in Section 4.1. Note that for simplicity the spectrochemical maps shown here only represent single-peak slice evaluations rather than full hyperspectral analysis, as we stated before. In difference to the raw-spectra shown in Figure 3 in Section 4.1, for the data discussed in this section, the spectra were corrected for fluorescence and stray light background prior to spectral slicing using the LARAsoft internal SCARF routine (see Ref. [16]). Afterwards, the treated data cube was evaluated for selected spectral slices, namely at the locations of the graphene peaks D, G, and 2D (at ~1350 cm^−1^, ~1600 cm^−1^, and ~2700 cm^−1^, respectively); and for reference at the Si/SiO_2_ peaks TO and 2TO (at 520 cm^−1^ and 970 cm^−1^, respectively).

As an example for spectrochemical mapping, the evaluation of a raster-scan of a GFET device (200 × 200 μm^2^, *Graphenea*) is shown in Figure 5. The sequence displays the same area segment of the device—indicated in panel (a) of the figure—with a series of Raman maps, utilizing a step size of ΔS = 2 steps (corresponding to spatial increments of 2.5 μm); note that examples for different step sizes are provided in the Appendix A. The raster maps shown here are for spectral slices for the three main graphene peaks. The following observations can be made:

First, the spectral raster maps clearly reveal the “edges” between the graphene chip and the Si/SiO_2_ substrate (or metal contacts); see panels (b) to (d).

Second, the spectral image maps for the different graphene peaks stem from the same raster-scan and thus are directly correlated. With the spatial step increment ΔS = 2 steps ≡ 2.5 μm (substantially smaller than the laser spot with FSD ≅ 7.1 μm), some “structural features” can be identified as being common to all three spectral images, although they become increasingly blurred in the sequence, due to the decreasing signal intensities—notably $I_{\mathrm{peak}-2D}>I_{\mathrm{peak}-G}\gg I_{\mathrm{peak}-D}$ −, and thus a noisier signal.

### 4.5. Comparison of Raman Raster Maps with Inages Obtained by Complementary Techniques

In order to ascertain that apparent spectro-spatial features could be associated with real spatial differences in the sample, we performed a series of test measurements in which we compared the spectrochemical raster maps with images of the same area obtained using different imaging techniques. 

In principle, this aspect is already beyond the scope of this publication. Nevertheless, for the interested reader, two examples are provided in the Appendix A, namely for a GFET device sample—SEM image versus Raman map (see Appendix A and for a Graphene-on-glass sample—wide-field image vs. Raman map (see Appendix A. As was pointed out in the introduction, one of the initial purposes of setting up this custom-built CRM was to be able to analyze graphene samples before, during, and after exposure to tritium or larger tritium-substituted molecules, to monitor chemical bonding changes in graphene, and to ascertain tritium loading. These two examples may be seen as precursor experiments to said goals.

## 5. Conclusions and Outlook

Within the framework of the current investigation, we built—and tested—two quasi- identical CRM systems, using off-the-shelf optomechanical and optical components; the setup followed the design ideas developed in our Raman laboratory at the Universidad Autónoma de Madrid (UAM). The two systems differed only in the type of 532 nm DPSS laser excitation source (fiber-coupled miniature laser vs. free-space launch laser module), and the motorized xyz-translation assembly for sample motion (driven by micro-stepper motors vs. piezoelectric inertia actuators), to demonstrate the versatility of the design. One of the systems is now used in routine Raman mapping analysis, while the other one serves to predominantly further development and test tasks.

During the now more than 18 months of regular use, our CRM systems have proven their reliability, and they fulfilled the remits stated in the introduction. The achievements from our development and test measurement campaigns can be summarized as following.

First, a near-optimal confocal configuration could be established, experimentally determined by axial Raman spectral scans through the laser focus position on a reference sample. For this, we utilized a series of pinholes of different diameters; then, the one was selected whose experimental data came closest to the diameter deduced from theoretical estimates (see Section 4.3). In the wake of these measurements, we experimentally confirmed the expected Rayleigh depth range as well (see Appendix A.

Second, all characterization and test sample measurements were carried out using the 10× microscope objective; note that in the current CRM configuration, a (lateral) spatial resolution of ~5 μm was achieved—based on the focal beam diameter (see Section 4.3) and the Rayleigh criterion for the optical separation of features. Currently, extended tests are under way, utilizing higher-magnification 20× and 40× objectives (the latter constituting the limit in terms of working distance for our present sample holder unit). Preliminary results indicate that the CRM performs as expected under higher magnification, too.

Third, we successfully performed Raman spectral raster scans of thin-film graphene samples. Different raster increments were trialed, ranging from less than the laser focal beam diameter (FBD) to substantially larger steps. In the former case, Raman raster maps revealed spatially-resolved details, such as, e.g., precise edges between different materials (see the results in Section 4.4), being consistent with theoretical expectations. In the latter case, we could demonstrate that in-focus scans could be performed across the full area of our standard samples (10 × 10 mm^2^).

Fourth, for a small set of samples, Raman raster images were compared with images obtained by other imaging methods, in this study scanning electron microscope (SEM) images and wide-field white-light CCD images. In both cases, compositional variation evident in the Raman maps could be matched to the same spatial structures observed in the chemical-unspecific images (see the results in Appendix A).

Fifth, we would like to note that, for the present study, only single spectral-slice data were utilized in the generation of Raman maps. Of course, this is insufficient to extract subtle composition variations or to identify different species with similar spectral fingerprints; multi-line or hyperspectral analysis (likely in conjunction with multivariate analysis) then becomes indispensable [30,31] to interpret, e.g., properties like the thickness of graphene layers [32,33]. We are in the process to begin incorporating associated algorithms into our data acquisition and analysis software suite for automated, species-specific Raman image generation.

We like to note that, in conjunction with the aims set at the outset of this study—i.e., the analysis of samples previously exposed to or even held under tritium gas—we prepared the required experimental environment. A dedicated sample chamber was tested and certified for tritium operation, and the CRM was transferred to the “hot” area of the tritium laboratory for its safe connection to the tritium-exhaust system. Besides the planned tritium loading of graphene samples, in pre-tests we had already established that spatially-resolved, chemical-specific Raman maps can be generated from samples akin to the rear-wall material used in the KATRIN experiment [34], albeit yet without exposure to tritium. During the latter stages of writing, said rear-wall materials were exposed to a T_2_/CT_4_ gas mixture for several days. After lengthy checks for safe handling, these samples were then transferred to our CRM for Raman raster-scanning. The recorded data are currently under evaluation to ascertain the presence of tritiated adsorbates; results should be published in the near future.

Finally, we would like to add a few general remarks in relation to setting up an in-house built CRM. 

In principle, the hardware itself can be assembled relatively quickly using off-the-shelf optical and optomechanical cage-system components. The only, (slightly) tricky part is the optical alignment of the CRM; despite the excellent precision inherent to the *Thorlabs* cage system, displacement/misalignment in the system of the order of a few micrometers is unavoidable. However, a reasonably skilled person can easily compensate for this, making use of the positional and angular adjusters of key optical components. Of course, incorporating such a CRM into a “hostile” environment—such as a tritium-compatible glove box, as intended in our studies—carries its own constructional overheads and requires careful planning and design.

Apart from the Raman excitation laser and the spectrometer—systems which are often already available in spectroscopy laboratories—the overall cost of the CRM device described here was only ~10 k€, including the motorized xyz-sample stage, but excluding the laser and spectrometer. Thus, such a cost-effective instrument may not only lend itself for the specialized R&D tasks addressed here, but could also be utilized (i) in many other experiments requiring good basic CRM capabilities; or (ii) to provide a modular system for, e.g., postgraduate training (to setup and learn about CRM properties for scientific spectroscopy projects).

## Figures and Tables

**Figure 1 sensors-22-10013-f001:**
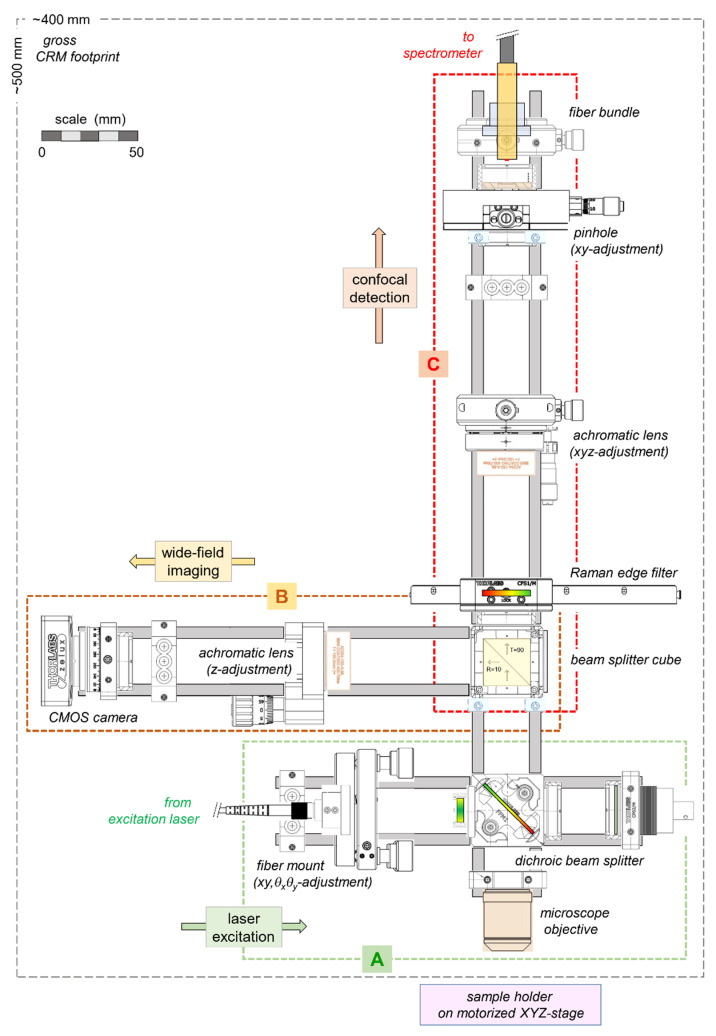
Overview of the confocal Raman microscope constructed from opto-mechanical cage system components and optical elements of the *Thorlabs* collection; note that the DPSS excitation laser, the Raman spectrometer, and the motorized sample stage are not included here. The conceptual CRM groups are: **A**—excitation laser coupling and Raman light collection; **B**—the wide-field imaging arm; **C**—the confocal light collection arm. Key optical components are annotated, together with their spatial adjustment; for further details, see main text.

**Figure 2 sensors-22-10013-f002:**
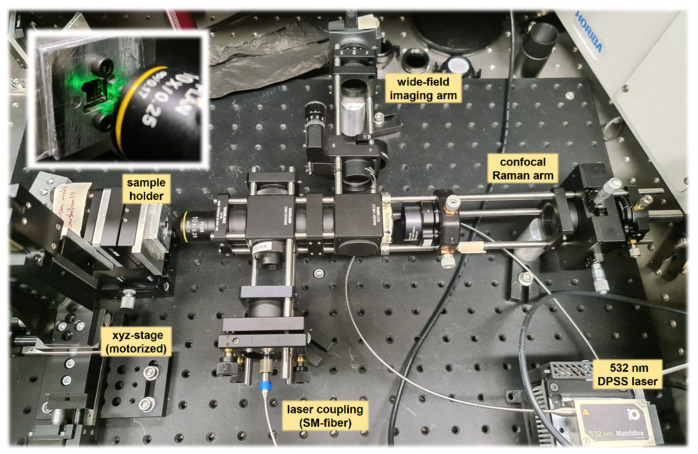
Overview image of the confocal Raman microscope system, as shown schematically in Figure 1, together with the xyz-motorized sample stage and the (fiber-coupled) excitation laser. Insert: a GFET sample mounted on the xyz-stage, illuminated by laser light through the CRM objective. Key structural elements in the image are annotated; breadboard tapped-hole spacing = 25 mm.

**Figure 3 sensors-22-10013-f003:**
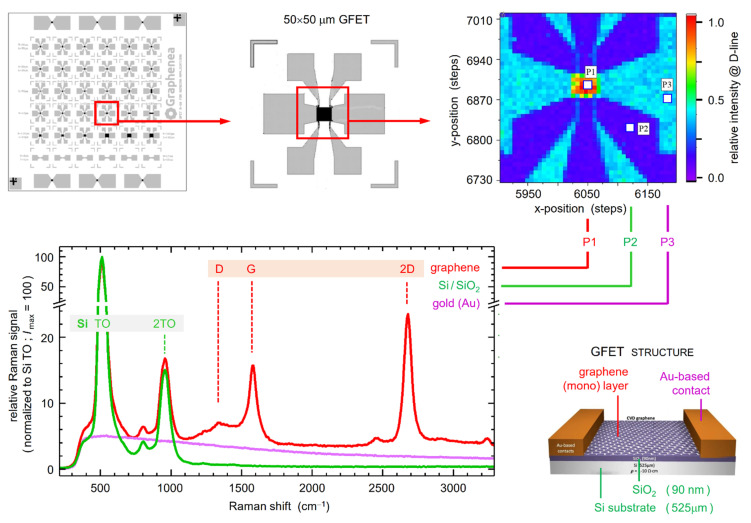
Concept of Raman spectral mapping, exemplified for a graphene sample (GFET-S10, *Graphenea*; image reproduced with permission), with the recorded data related to the indicated 50 × 50 μm^2^ GFET device. The area is raster-scanned (here with a step-increment of ΔS = 7 (equivalent to ~9 μm)), with a sequence of full spectra recorded at each raster point. Out of the hyperspectral stack, a particular slice is selected to generate a Raman map of the sample (here for the data at about 1330 cm^−1^). Selected spectra from three specific locations in the GFET structure—graphene layer, gold-contact, and substrate (marked by the white squares)—are displayed in the lower left, highlighting the distinct spectral differences associated with the (chemical) sample constituents. For further details, see text.

**Figure 4 sensors-22-10013-f004:**
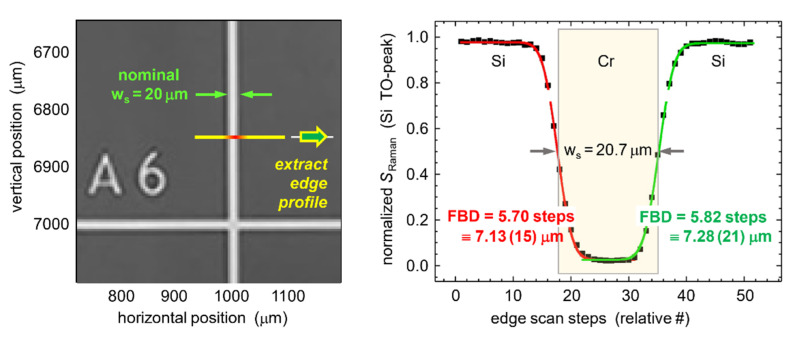
Determination of the laser focal beam diameter (= FBD) on the target surface, through the ×10 microscope objective. Line-scan for the focused laser beam across a chromium grid line of a silicon SEM-finder grid substrate (EM-Tec FG1, *Micro to Nano*), using the edge-scan method [25]; the scan data are for the 2TO spectral line of silicon. For details, see text.

**Figure 5 sensors-22-10013-f005:**
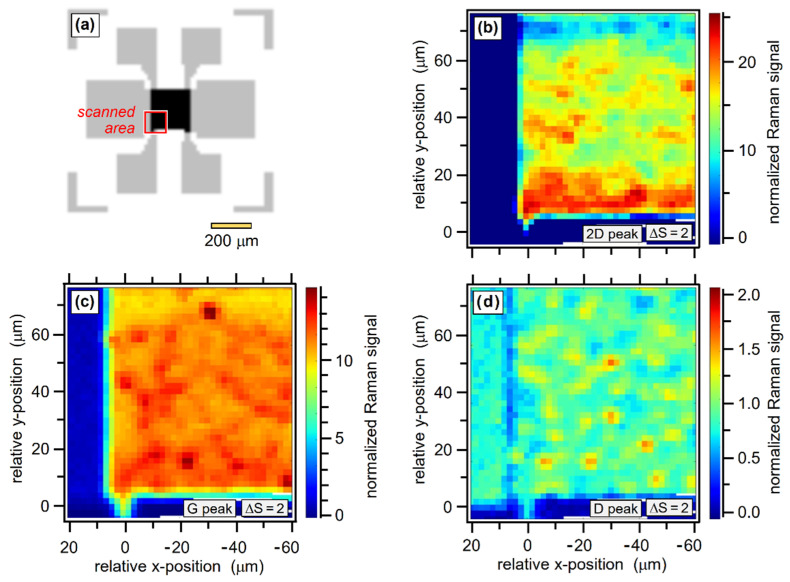
Raman image maps of a GFET device (200 × 200 μm^2^, *Graphenea*). (**a**) Schematic GFET structure; the area scanned by CRM—about 80 × 80 μm^2^—is indicated by the red square; the spatial reference point for the scans is the left-most contact. (**b**–**d**) Maps of the 2D-, G- and D-peak signal, respectively, scanned with a step increment of ΔS = 2. Note that 1 step-increment (ΔS = 1) corresponds to a spatial displacement of ~1.25 μm. For further details, see text.

**Table 1 sensors-22-10013-t001:** Specification of key optomechanical and optical components used in the current confocal Raman microscope (CRM) setup.

Component	Part Number	Remarks
Fixed-focus collimator	F260FC-A, *Thorlabs*	FC-coupled fiber collimator ^(1)^
Laser beam steering	KC1-S/M, *Thorlabs*	$x,y,\theta_{x},\theta_{y}$ kinematic mount
Laser line filter	LL01-532-12.5, *Semrock*	Laser clean-up filter
Laser beam separator	Di02-R532-25×36, *Semrock*	Single-edge dichroic beam splitter
Microscope objective	LIO-10X, *Newport/MKS*	10× infinity-corrected objective ^(2)^
Laser power monitor	SM1PD1A, *Thorlabs*	SM1-mounted Si photodiode
Beam splitter cube	BS025, *Thorlabs*	Non-polarizing, ratio 10:90 (R:T)
Focusing lens	AC254-150-A-ML, *Thorlabs*	AR-coated achromat, f = 150 mm
Color camera	CS165CU/M, *Thorlabs*	Color CMOS camera, USB2
Raman edge filter	LP03-532RU-25, *Semrock*	Long-pass 532 nm edge filter ^(3)^
Focusing lens	AC254-150-A-ML, *Thorlabs*	AR-coated achromat, f = 150 mm
Confocal pinhole	PxxK, *Thorlabs*	SS-foil pinhole, xx = ∅ in μm ^(4)^
Pinhole xy-adjustment	ST1XY-S/M, *Thorlabs*	xy-translator (micrometer drive)
Spectrometer fiber bundle	Custom-made, *CeramOptec*	48-fiber bundle, circular-to-slit ^(5)^

^(1)^ Alternative collimation package may be selected, to provide different laser beam waist. ^(2)^ Objectives with different magnification or of different type, compatible with 30 mm cage mounting, may be used. ^(3)^ This Raman edge filter is optional, to remove residual 532 nm laser radiation transmitted through the dichroic laser beam splitter. ^(4)^ The pinhole diameter used in this study was 75 (or 100) μm; for higher-magnification objectives the diameter needs to be adapted for confocality. ^(5)^ The fiber coupling between the CRM and spectrometer may need to be adapted, in case a different spectrometer is used.

**Table 2 sensors-22-10013-t002:** Operational data for the linear translation stages used in this study. (1) xyz-assembly built with encoded stepper-motor-driven stages 8MT173-20-MEn1 (*Standa*); driver: three-axis controller 8SMC5-USB-B8-B9 (*Standa*); and (2) xyz-assembly based on piezo-inertia translators (xy-direction—PD1/M Open-Loop stages (*Thorlabs*); z-direction—PIA25 Open-Loop drive and LX20/M stage (*Thorlabs*); driver: four-channel K-cube controller KIM101 (*Thorlabs*).

Parameter	8MT173-20-MEn1	PD1/M	PIA25 (+LX20/M)
Travel range	20 mm	20 mm	25 mm
Drive	Stepper motor	Piezo inertia	Piezo inertia
Position encoder	Yes	No	No
Resolution, full-step	1.25 μm	1 μm ^(3)^	20 nm ^(3)^
Resolution, micro-step	0.156 μm (1/8 step) ^(1)^	NA	NA
Backlash	~2 steps ^(2)^	None	None
Speed, continuous stepping	5 mm/s	3 mm/s	2 mm/min

^(1)^ Micro-stepping control is possible down to 1/256 of full-step; in this work, it was limited to less than 1/8 full-step. ^(2)^ To eliminate forward/backward positional ambiguity, translation advance is unidirectional from a reference point (when operated in *Backlash compensation mode*). ^(3)^ This value may vary by up to 20% (associated with component variance, change of direction, and application conditions); it is adjustable by about 30%, changing the piezo actuator’s operating parameters.

## Data Availability

Selected experimental data and parts lists are available from the authors on request.

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
