# Peer review of "Versatile Confocal Raman Imaging Microscope Built from Off-the-Shelf Opto-Mechanical Components"

_sensors, 2022, doi:10.3390/s222410013_

Round 1
Reviewer 1 Report
In this paper the authors provide a detailed overview of the construction of a confocal Raman microscope. They list the components and detailed description of the alignment procedure as well as some test of the instrument performance (based in graphene samples).
The manuscript is clearly written and easy to follow, the english is clear and accurate; however, my main critisism is that the manuscript is too long. Some specific comments:
1. Some ideas are repeated more than once, although they migth be important, given the length of the mauscript this should be aovided. Some explanaitons are too long, shorten specially the ones that are not general in applicability or that important for the actual construction/alignment/evaluation of the microscope. One example of this is the full discussion and examples around the knife edge techinque and the edges of the different sample regions; in my opinion it is fine to mention it, ilustrate its use once and give references where the interested reader can find the details. This part takes around two pages. Another part that feels of not so much use is the extensive discussion around the registration of SEM and Raman images; this is an important point but, for a paper which its main goal is to present the construction of the microscope, I find the discusson too long and not very useful.
2. It is my impression that a good part of the potential readers would come to the paper from a Raman background (obviously, I am not sure of this but it seems to me the most natural target readership). If this is true, they might not have an uderstanding of the principles of confocal microscopy. I think the authors should provide at least a brief (one or two paragraphs) introduction to the topic as well as specialized references for the interested reader.
3. Although this number will depend on the sample, available spectrometer and detector, it would be nice to see some typical acquisition times for the spectra and imaging. The paper mentions them but it would be better to metion this explicitly for a few cases.
4. The paper gives the impression that the magnification of the objective is the critical parameter for resolution, pinhole selection, etc. This is not true, the relevant parameter is the numerical aperture of the objective, and as mentioned in the text, the effective NA if the back aperture of the objective is not being overfilled.
5. The proposed angular correction for the sample position discussed around equations (1) and (2) is important; however given the very long Raleigh range of the 10X objective that the authors use, I think it is almost pointless to evaluate this parameter. This is because the Raleigh range will allow for a deviation of a few degrees; a much more useful case would be if the authors can test their proposed correction with a high NA objective with a significantly smaller NA.
6. Although Figure 2 gives a full view of the instrument, it could be useful for the authors to provide the approximate dimensions of the whole setup.
Author Response
see attached pdf-file

Reviewer 2 Report
In the submitted manuscript, Barrero et al presented a detailed instruction and helpful guidance to build a confocal Raman microscope using off-the-shelf components. I think that this work will be very useful for those who want to build a confocal Raman microscope in their lab, assuming an excitation laser and a spectrometer are already available. The manuscript is very well written, and the detailed explanation of the alignment and testing procedures is very informative. My major concern is that the potential advantages of the proposed system over a commercial system have not been clearly justified.
1. The authors claim the proposed system is more suitable for an application that involves radioactive or toxic materials. Is it because the modular design would allow a user to replace a contaminated part at a lower price? Noteworthy, most radioactive materials including tritium can be cleaned. The time and effort required to clean a spill may be less than those required to replace the contaminated part and realign the system.
2. The authors also claim that the proposed system would be better for an operation in a hostile environment. However, the confocal microscope, which uses a point scan, requires a stable environment irrespective of the implementation. In my opinion, the proposed system is quite susceptible to the environmental noise, as the microscope and the sample are independently mounted. I agree the developed system could be placed in a glove box more easily for the measurement of radioactive samples; however, this capability has not been demonstrated in the submitted manuscript.
3. The price advantage is a bit misleading. An excitation laser and a spectrometer are the most expensive items of a commercial confocal Raman microscope.
Author Response
see attached pdf-file

Round 2
Reviewer 2 Report
I am still not convinced about the scientific merits of the proposed system and its potential advantages over a commercial system; however, I think the information contained in the submitted manuscript will benefit the readers who are interested in building a confocal Raman microscope in their lab. Thus, I recommend its publication in Sensors.